# A conserved bacterial protein induces pancreatic beta cell expansion during zebrafish development

Jennifer Hampton Hill[1], Eric A Franzosa[2,3], Curtis Huttenhower[2,3], Karen Guillemin[1,4]*

[1]Institute of Molecular Biology, University of Oregon, Eugene, United States; [2]Biostatistics Department, Harvard T. H. Chan School of Public Health, Boston, United States; [3]The Broad Institute, Cambridge, United States; [4]Humans and the Microbiome Program, Canadian Institute for Advanced Research, Toronto, Canada

**Abstract** Resident microbes play important roles in the development of the gastrointestinal tract, but their influence on other digestive organs is less well explored. Using the gnotobiotic zebrafish, we discovered that the normal expansion of the pancreatic β cell population during early larval development requires the intestinal microbiota and that specific bacterial members can restore normal β cell numbers. These bacteria share a gene that encodes a previously undescribed protein, named herein BefA (β Cell Expansion Factor A), which is sufficient to induce β cell proliferation in developing zebrafish larvae. Homologs of BefA are present in several human-associated bacterial species, and we show that they have conserved capacity to stimulate β cell proliferation in larval zebrafish. Our findings highlight a role for the microbiota in early pancreatic β cell development and suggest a possible basis for the association between low diversity childhood fecal microbiota and increased diabetes risk.

*For correspondence: kguillem@ uoregon.edu

**Competing interests:** The authors declare that no competing interests exist.

## Introduction

Host-associated microbes play important roles in the development of animal digestive tracts (*Bates et al., 2006*; *Semova et al., 2012*; *Sommer and Bäckhed, 2013*). Using the gnotobiotic zebrafish model, our group has shown previously that resident microbes promote host processes in the developing intestine such as epithelial differentiation (*Bates et al., 2006*) and proliferation (*Cheesman et al., 2011*). The role of microbes in the development of other digestive organs remains underexplored, despite the fact that many diseases in peripheral digestive organs are correlated with microbial dysbiosis (*Chang and Lin, 2016*; *Gülden et al., 2015*). The ability to manipulate resident microbes in the larval zebrafish (*Milligan-Myhre et al., 2011*), combined with the optical transparency and sophisticated genetic tools of the zebrafish model, make it a powerful platform to investigate this question. Here, we use gnotobiotic zebrafish to demonstrate a role for resident microbes in promoting pancreatic β cell development.

The zebrafish has a well-characterized program of β cell development, which is highly conserved with that of mammals (*Kinkel and Prince, 2009*). In the zebrafish embryo, initial β cells arise from precursors within the dorsal and ventral pancreatic buds (*Biemar et al., 2001*; *Field et al., 2003*; *Wang et al., 2011*). The two buds fuse by 52 hr post fertilization (hpf), and give rise to the fully fated pancreas with only a single islet of hormone-secreting endocrine cells, by 3 days post fertilization (dpf) (*Biemar et al., 2001*; *Field et al., 2003*; *Kumar, 2003*). Coinciding with the approximate time of larval emergence from the chorion by 3 dpf, these newly fated β cells begin to expand (*Chung et al., 2010*; *Dong et al., 2007*; *Hesselson et al., 2009*; *Kimmel et al., 2011*; *Moro et al.,*

**eLife digest** For a long time, genes and carefully orchestrated chemical signals from the mother's body have been known to shape the earliest phases of an embryo's development. More recently, researchers have started to appreciate that environmental cues – such as signals from nearby bacteria – also shape animal development.

The body is home to a teeming community of bacteria and other microbes, called the microbiota. Because the gut houses more microbes than any other part of the body, the role that the microbiota plays in gut development has been investigated. Less is known about how the microbiota affects the development of the other organs involved in digestion, such as the pancreas. In developing fish and mammals, the pancreas grows its population of insulin-producing beta cells during the same period of development in which the microbial population of the gut becomes established.

Now, Hill et al. show that certain gut bacteria are necessary for the pancreas to populate itself with a robust number of beta cells during development. Normally, the number of beta cells in zebrafish larvae increases steadily in the first few days after hatching. However, developing zebrafish that were reared in a microbe-free environment maintained the same number of beta cells as they had before hatching. Exposing the microbe-free fish to certain bacteria restored their beta cell populations to normal levels. Further investigation revealed that these bacteria release a protein called BefA that causes the beta cells to multiply.

Some bacteria in humans produce proteins that are similar to BefA. Hill et al. performed experiments that showed that these proteins also stimulate beta cell development in microbe-free fish. Future studies are now needed to investigate the mechanism by which the proteins affect beta cell development, and to find out whether they have the same effect in humans and other animals. This will help us to understand whether a lack of gut microbes could contribute to the development of diseases, such as diabetes, that are characterized by insufficient numbers of beta cells.

*2009*). β cells derived from the dorsal bud become quiescent, while ventral bud derived β cells begin to undergo expansion via mechanisms of both proliferation and neogenesis (*Hesselson et al., 2009*). Between 3 and 6 dpf, the number of β cells within the primary islet will almost double (*Moro et al., 2009*). Intestinal colonization with microbes occurs concurrently with this early larval period of β cell expansion. Following development of the gut tube within the sterile embryo, the intestine of the emergent larva becomes open to the environment at both the mouth and the vent by 3.5 dpf, allowing for inoculation by environmental microbes (*Bates et al., 2006*). Within the larval gut, bacteria proliferate rapidly, such that a single species in mono-association can reach the luminal carrying capacity within several hours (*Jemielita et al., 2014*).

Human post-natal β cell expansion also occurs concurrently with intestinal tract colonization by commensal microbes. In utero, β cells are produced via differentiation from progenitors (*Georgia et al., 2006*; *Stanger et al., 2007*) and at birth this newly fated cell population begins to expand by self-proliferation (*Georgia and Bhushan, 2004*; *Gregg et al., 2012*; *Kassem et al., 2000*; *Teta et al., 2007*). β cell proliferation rates peak at 2 years of age and then steadily decline (*Gregg et al., 2012*). By 5 years of age, most of the β cell mass has become slow cycling and will not expand significantly again unless stimulated by elevated metabolic demands, such as obesity or pregnancy. At birth, infants are exposed to their mothers' vaginal, fecal and skin associated microbes, which immediately begin to colonize the neonatal intestine (*Biasucci et al., 2010*; *Dominguez-Bello et al., 2010*; *Palmer et al., 2007*). By 3 years of age, the composition and complexity of the microbiota typically resembles that of an adult associated community (*Murgas Torrazza and Neu, 2011*; *Palmer et al., 2007*; *Yatsunenko et al., 2012*). However, factors such as diet, birth mode and antibiotic exposure can result in reduced microbial taxonomic diversity during these early years of life (*Mueller et al., 2015*). Notably, factors that reduce microbiota diversity are also associated with increased risk for diabetes mellitus (*Knip et al., 2005*). Loss of β cell function through autoimmunity results in abnormal glucose homeostasis and is the cause of type 1 diabetes (T1D) in humans. Recent studies have shown that decreased taxonomic diversity of the intestinal microbiota

is correlated with T1D (*Brown et al., 2011*; *Giongo et al., 2011*). Indeed, loss of bacterial diversity precedes the onset of T1D in children, and may play a causative role in disease (*Kostic et al., 2015*).

To our knowledge, no one has yet investigated a role for the gut microbiota in the development of pancreatic β cells. Communication between the intestine and the pancreas is critical for overall homeostasis. The two organs are therefore connected physically, metabolically, and developmentally in order to carry out their essential functions. We propose that this established and important connection might also mediate the influence of resident microbes on developmental processes in the pancreas. Here we examine the effects of microbial colonization on initial expansion of zebrafish primary islet β cells. We find that β cell mass expansion, up to at least 6 dpf, is promoted by the presence of the microbiota. Using a culture collection of zebrafish intestinal bacteria, we show that certain strains can restore β cell expansion in germ free (GF) fish. We report the discovery of a secreted protein, shared among these strains and named herein β cell expansion factor A (BefA) that is sufficient to recapitulate this effect. Homologs of the *befA* gene are present in the genomes of a subset of human intestinal bacteria, and we show that two of the corresponding proteins share BefA's capacity to induce β cell expansion in zebrafish.

## Results

### The microbiota is required for normal expansion of the larval β cell mass

To investigate a possible role for the microbiota in pancreas development and specifically in β cell expansion, we quantified total β cells in GF and conventionally reared (CV) *Tg(-1.0insulin:eGFP)* fish (*diIorio et al., 2002*) at 3, 4, 5 and 6 dpf (*Figure 1A*, *Figure 1—source data 1*). The number of β cells in CV fish increased steadily from 3 to 6 dpf (*Figure 1A*). However, the average number of β cells in GF fish remained static over this time (*Figure 1A*). Furthermore, at 6 dpf, the overall structure of β cells within the primary islet also appeared much less densely packed in GF than in CV fish (*Figure 1B*). This effect is not likely to be due to changes in initial differentiation of the β cell population since the total number of β cells is not different between GF and CV fish at 3 dpf (*Figure 1A*), a time at which exposure to bacteria is also limited.

Because insulin from β cells functions to reduce levels of circulating glucose, we tested whether the β cell deficiency in GF larvae at 6 dpf affected the metabolic function of the fish by measuring free glucose levels. The amount of glucose detected in GF fish was significantly higher than in CV fish (*Figure 1C*, *Figure 1—source data 2*). These data suggest that GF fish, with a paucity of β cells, are less efficient at importing and processing glucose from the blood due to lower levels of circulating insulin. This is consistent with previous studies showing free glucose levels in zebrafish larvae to be correlated with β cell numbers (*Andersson et al., 2012*).

### Only specific bacterial members of the zebrafish microbiota are sufficient to rescue normal expansion of the GF β cell mass

We developed an experimental timeline, depicted in *Figure 2A*, to test the capacity of individual zebrafish bacterial isolates to induce β cell expansion. We derived embryos GF at 0 dpf and allowed them to develop in this environment until after hatching. At 4 dpf, when the GF larvae have a patent gut tube, we inoculated them with defined microbes and/or microbial derived products by adding these directly to the embryo media. The fish were incubated with the treatment of interest for 48 hr before analysis of the β cell mass at 6 dpf.

We found that we could rescue β cell numbers to CV levels by the addition of non-sterile, normal fish tank water to GF larvae at 4 dpf (*Figure 2B*, *Figure 2—source data 1*), suggesting that development of the normal number β cells is dependent upon microbes or microbial-derived products present in the water. We next inoculated 4 dpf GF larvae with a selection of bacterial isolates from the zebrafish intestine (*Stephens et al., 2015*) as well as one other related strain (*Bomar et al., 2013*). We prioritized bacterial strains that were capable of forming robust mono-associations with larvae between 4 and 6 dpf, as measured by the number of bacteria found within the gut at 6 dpf (*Figure 2C*). We found that the mono-associations with three different species of the genus *Aeromonas* and one species of the genus *Shewanella* was sufficient to rescue GF β cell numbers to levels observed in CV fish (*Figure 2B*, *Figure 2—source data 1*). Importantly, other isolates such as *Vibrio*

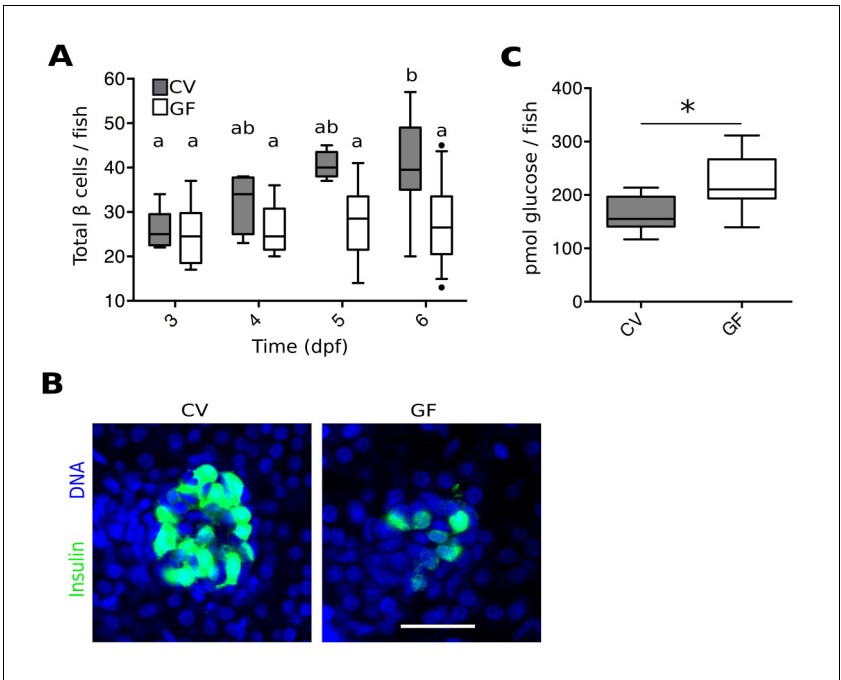

**Figure 1.** The microbiota are required for normal expansion of the larval β cell mass. (**A**) Total number of β cells per larva in GF (white box plots) and CV (grey box plots) fish at 3, 4, 5 and 6 dpf. In this, and in all subsequent figures, CV data are shown in grey box plots, and GF data, or statistically similar treatment groups, are shown in white box plots. In all relevant panels and remaining figures, box plot whiskers represent the 95% confidence interval of the data set. Single factor ANOVA indicates that gnotobiology of the fish was significant in determining the number of β cells present ($F_7$=9.01, p=1.45e$^{-8}$). Labels a, ab and b indicate the results of post hoc means testing (Tukey). The difference between GF and CV cell counts became significant at 6 dpf (t=−5.91, p<0.001). (**B**) Representative 2D slices from confocal scans through the primary islets of 6 dpf CV and GF *Tg(-1.0insulin:eGFP)* larvae. Each slice is taken from the approximate center of the islet structure. Insulin promoter expressing β cells are in green and nuclei are blue. Scale bar = 40 µM. (**C**) The average amount of glucose (pmol) per larva aged 6 dpf (* $t_{17}$=−3.65, p<0.01).

The following source data is available for figure 1:

**Source data 1.** Quantifications and statistical analysis of larval β cells corresponding to *Figure 1A*.
**Source data 2.** Quantifications and statistical analysis of 6 dpf larval free glucose levels corresponding to *Figure 1C*.

---

*sp.* and *Delftia sp.* were not sufficient to rescue this phenotype (*Figure 2B*, *Figure 2—source data 1*), indicating that only specific members of the microbiota are capable of inducing expansion of the β cell mass.

### *Aeromonas* secretes a factor that rescues normal expansion of the GF β cell mass

Bacterial interactions with host organisms often involve secreted molecules. To test whether a secreted bacterial factor(s) could influence β cell expansion, we harvested cell free supernatant (CFS) from overnight cultures of each *Aeromonas* strain shown to rescue β cell expansion (*Figure 2B*) and added these to GF larvae at 4 dpf. For each of the three strains of *Aeromonas* tested, the CFS alone was able to restore β cell numbers in GF fish (*Figure 3A*, *Figure 3—source data 1*), indicating that a secreted factor (or factors) produced by these bacteria is (are) sufficient to induce β cell expansion. As a control, we also treated GF fish with CFS from a *Vibrio sp.* isolate, which colonized the zebrafish gut (*Figure 2C*, *), but did not induce β cell expansion (*Figure 2B*, *). We found the number of β cells in fish receiving *Vibrio* CFS was not significantly different from that of GF fish (*Figure 3A*, *Figure 3—source data 1*). Furthermore, the capacity to induce increased β cell numbers was lost when

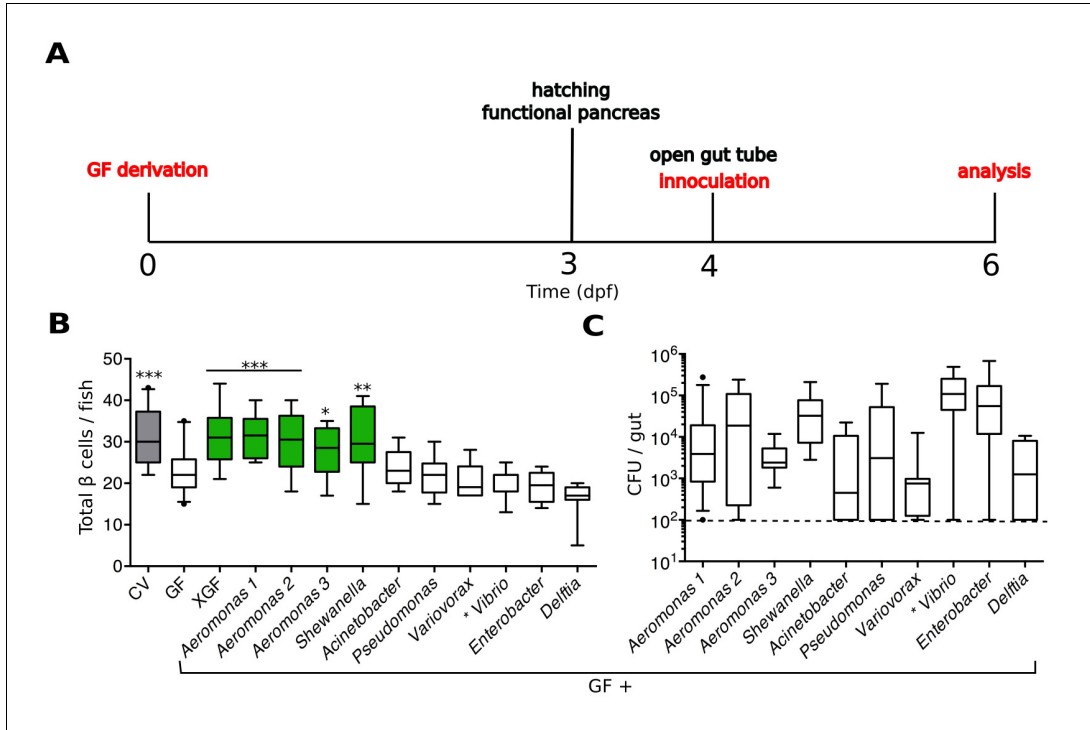

**Figure 2.** Specific bacterial members of the zebrafish microbiota are sufficient to rescue normal expansion of the GF β cell mass. (A) Experimental timeline for all subsequent zebrafish experiments, unless stated otherwise. Experimental manipulations are denoted by red text. Important zebrafish developmental events are denoted by black text. (B) Quantification of β cells in CV, GF and GF larvae treated at 4 dpf with either non-sterile tank water (XGF) or mono-associated with a specific bacterial strain. Bacterial mono-associations are labeled by genus. Different *Aeromonas sp* are labeled with a number (1, 2 or 3). *p<0.05, **p<0.01, ***p<0.001: Denotes treatment that is significantly different than GF by Tukey analysis. Additionally, here and in all subsequent figures, significant data sets (p<0.05 when compared to GF) are also highlighted as green box plots. (C) Bacterial isolates of the zebrafish gut and related strains are capable of forming mono-associations with larvae from 4 to 6 dpf. Quantification of the colony forming units (CFUs) per gut for each bacterial strain, assayed after 48-hr exposure to GF larvae. Dashed line denotes the limit of detection.

The following source data is available for figure 2:

**Source data 1.** Quantifications and statistical analysis of larval β cells corresponding to *Figure 2B*.

the *Aeromonas 1 (A. veronii)* CFS sample was treated with proteinase K (*Figure 3A*, *Figure 3— source data 1*), indicating that our secreted factor(s) of interest was likely to be a protein. Because of existing genetic reagents available for the *A. veronii* strain (*Bomar et al., 2013*), and its capacity to modulate traits of gnotobiotic zebrafish and other hosts (*Bates et al., 2006*; *Cheesman et al., 2011*; *Graf, 1999*; *Rolig et al., 2015*), we focused on this strain for the remainder of our analysis.

To narrow down the list of candidate proteins secreted by *A. veronii*, we tested whether the activity was present in the CFS of an *A. veronii*$^{ΔT2SS}$ mutant strain (*Maltz and Graf, 2011*) lacking a functional type 2 secretion system (T2SS), one of the major protein secretion pathways of Gram-negative bacteria. Despite the fact that it has a reduced secretome, CFS harvested from the *A. veronii*$^{ΔT2SS}$ strain was sufficient to rescue GF β cell numbers (*Figure 3A*, *Figure 3—source data 1*). Conveniently, this finding significantly decreased the number of candidate secreted *A. veronii* proteins with β cell expansion capacity. This result also suggested that our protein(s) of interest was secreted through an alternative mechanism.

We next used ammonium sulfate precipitation to further separate proteins within the *A. veronii*$^{ΔT2SS}$ CFS. Each of the fractions was able to increase β cells in GF fish (*Figure 3B*, *Figure 3— source data 2*), suggesting that either *A. veronii*$^{ΔT2SS}$ produces multiple proteins with this activity, or

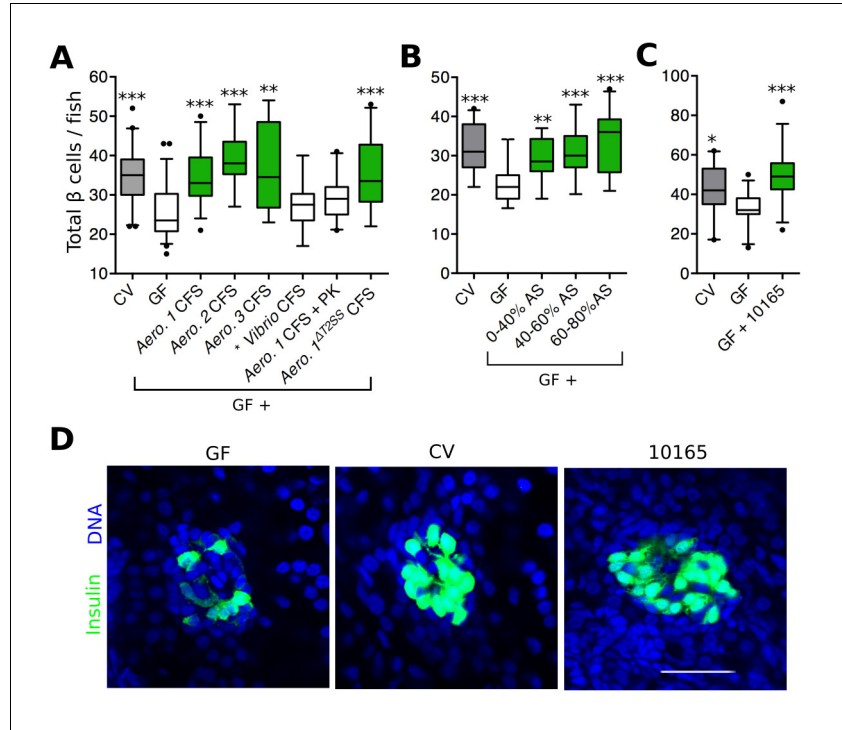

**Figure 3.** *Aeromonas* secretes a factor that rescues normal expansion of the GF β cell mass. (**A**) Total β cell numbers in GF, CV and GF fish treated at 4 dpf with different cell free supernatant (CFS) samples. 'Aero.' refers to bacteria of the genus *Aeromonas*, with each number (1, 2, 3) denoting a separate species. '+ PK' indicates proteinase K addition to the CFS sample prior to treatment. *p<0.05, **p<0.01, ***p<0.001: Denotes treatment that is significantly different than GF by Tukey analysis. (**B**) Total β cell numbers in CV, GF and GF fish treated at 4 dpf with separate ammonium sulfate fractions (% AS) prepared from the *Aeromonas 1^{ΔT2SS}* CFS. Note that the 60–80% ammonium sulfate fraction resulted in the greatest increase in β cell numbers. (**C**) Total β cells in GF, CV and GF fish treated with purified protein. 10165 represents purified protein from the *M001_10165* locus. (**D**) Representative 2D slices from confocal scans through the primary islets of GF, CV and 10165 protein treated *Tg(-1.0insulin:eGFP)* 6 dpf larvae. Insulin promoter expressing β cells are shown in green and nuclei are blue. Scale bar = 40 μM.

The following source data and figure supplement are available for figure 3:

**Source data 1.** Quantifications and statistical analysis of larval β cells corresponding to *Figures 3A*.

**Source data 2.** Quantifications and statistical analysis of larval β cells corresponding to *Figure 3B*.

**Source data 3.** Quantifications and statistical analysis of larval β cells corresponding to *Figure 3C*.

**Figure supplement 1.** 10165 (BefA) protein purification.

that the effector was present to some extent within each fraction. Since the 60–80% fraction was able to induce the greatest increase in β cell numbers (*Figure 3B*), we used mass spectrometry to analyze the content of this fraction, which led to the identification of 163 proteins (*Supplementary file 1*). To identify promising candidates from this list, we took advantage of the fact that our zebrafish-associated bacterial isolates, for which we have drafted genome sequences (*Stephens et al., 2015*), differed in their capacity to induce β cells (*Figure 2B*). Using basic local alignment search tool (BLAST) we identified those proteins from our candidate list that were, first, predicted to be encoded by the genomes of the four bacterial strains with β cell expansion capacity, and second, absent from the strains lacking this capacity. Our analysis identified one single candidate gene, denoted by the locus tag, *M001_10165 (10165)*, predicted to encode a putative protein

of 261 amino acids. Consistent with the candidate protein being found in the CFS, the putative protein contained a predicted N-terminal secretion sequence.

To test whether *10165* encoded the secreted protein responsible for inducing β cell expansion, we cloned the gene into an inducible expression vector in *E. coli* strain BL21, which contains no *10165* homologues in its genome. We expressed and purified the 10165 protein to homogeneity, as confirmed by SDS-page gel electrophoresis (*Figure 3—figure supplement 1*). Purified protein was added to flasks of 4 dpf GF zebrafish larvae. This treatment was sufficient to rescue β cell numbers to CV levels by 6 dpf (*Figure 3C*, *Figure 3—source data 3*). The islets of larvae treated with the purified protein were visibly expanded compared to those of GF animals (*Figure 3D*). Therefore, we have named this protein β cell expansion factor A (BefA) after its observed activity in zebrafish.

## *BefA* is required for *Aeromonas* to induce GF β cell expansion

To determine whether the *befA (10165)* locus is necessary for *A. veronii* to induce an increase in β cell numbers, we generated an *A. veronii*^ΔbefA mutant strain by replacing the coding region of *befA* with a chloramphenicol resistance gene. To ensure that the loss of the *befA* gene would not affect the ability of *A. veronii* to form mono-associations with larvae, we performed growth and colonization assays and saw no deficiency in either the in vitro growth rate (*Figure 4—figure supplement 1A*) or the ability of *A. veronii*^ΔbefA to colonize the GF intestine compared to the wild-type (WT) strain (*Figure 4A*). However, when inoculated in a 1:1 ratio together with *A. veronii*^WT, the *A. veronii*^ΔbefA strain showed a small yet reproducible fitness disadvantage as measured by colonization level and competition index after 48 hr (*Figure 4—figure supplement 1B,C*). This result indicates that BefA confers some colonization benefit for *A. veronii* within the larval gut.

GF fish were mono-associated with the *A. veronii*^ΔbefA strain, or treated with its CFS from 4 to 6 dpf. Neither treatment was sufficient to rescue β cell numbers to CV levels (*Figure 4B*, *Figure 4—source data 1*). However, mono-associations of *A. veronii*^ΔbefA could be complemented in trans with either CFS from *A. veronii*^WT or purified BefA protein, which resulted in the restoration of the β cell population (*Figure 4B*, *Figure 4—source data 1*). Taken together, these data demonstrate that the

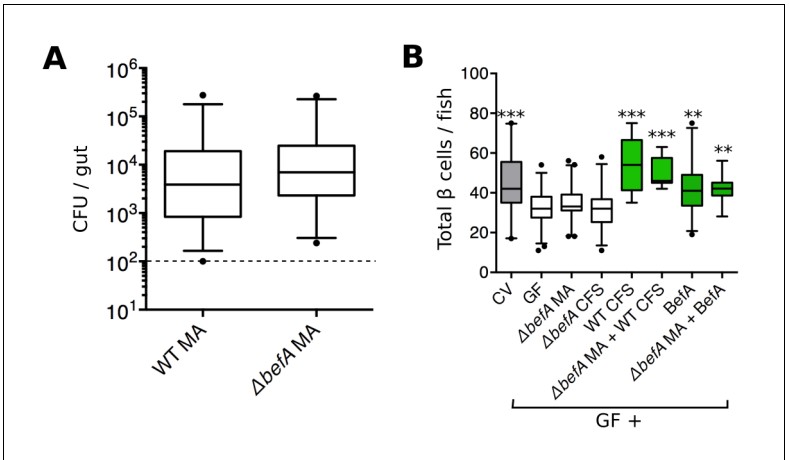

**Figure 4.** BefA is required for *Aeromonas* to induce GF β cell expansion. (**A**) Quantification of the colony forming units (CFUs) per gut in GF fish mono-associated (MA) with either wild type (WT) or mutant (△befA) A. veronii strains for 48 hr. Dashed line denotes the limit of detection (**B**) Total β cells in GF fish that have been mono-associated with △befA, treated with CFS from either WT or △befA, treated with purified BefA, or have been inoculated with a combination of these. **p<0.01, ***p<0.001: Denotes treatment that is significantly different than GF by Tukey analysis.

The following source data and figure supplement are available for figure 4:

**Source data 1.** Quantifications and statistical analysis of larval β cells corresponding to *Figure 4B*.

**Figure supplement 1.** *BefA* confers a colonization advantage in the larval zebrafish gut.

BefA protein is necessary in an *A. veronii* mono-association for early β cell expansion and suggests that *A. veronii* only produces a single effector of host β cell expansion.

## BefA facilitates β cell expansion by inducing proliferation

Proliferation is the primary mode of human neonatal β cell expansion (*Gregg et al., 2012*; *Kassem et al., 2000*; *Teta et al., 2007*). In 4–6 dpf zebrafish larvae, proliferation also contributes to β cell expansion (*Field et al., 2003*; *Hesselson et al., 2009*; *Moro et al., 2009*). Therefore, we investigated whether CV larvae had higher levels of β cell proliferation than GF larvae. 4 dpf larvae were treated with the thymadine analog, 5-ethynyl-2'-deoxyuridine (EdU) for 48 hr to mark cells that underwent proliferation during this time window. We found that, by 6 dpf, CV larvae had significantly more EdU labeled insulin-expressing cells than GF larvae (*Figure 5A,B*, *Figure 5—source data 1*). Next we asked whether treatment of GF larvae with BefA was sufficient to restore β cell proliferation to CV levels. We found that BefA-treated GF larvae had EdU incorporation similar to CV fish and significantly greater than GF (*Figure 5A,B*, *Figure 5—source data 1*). CFS from our *A. veronii*^ΔbefA strain was not sufficient to increase proliferation rates in GF fish (*Figure 5B*, *Figure 5—*

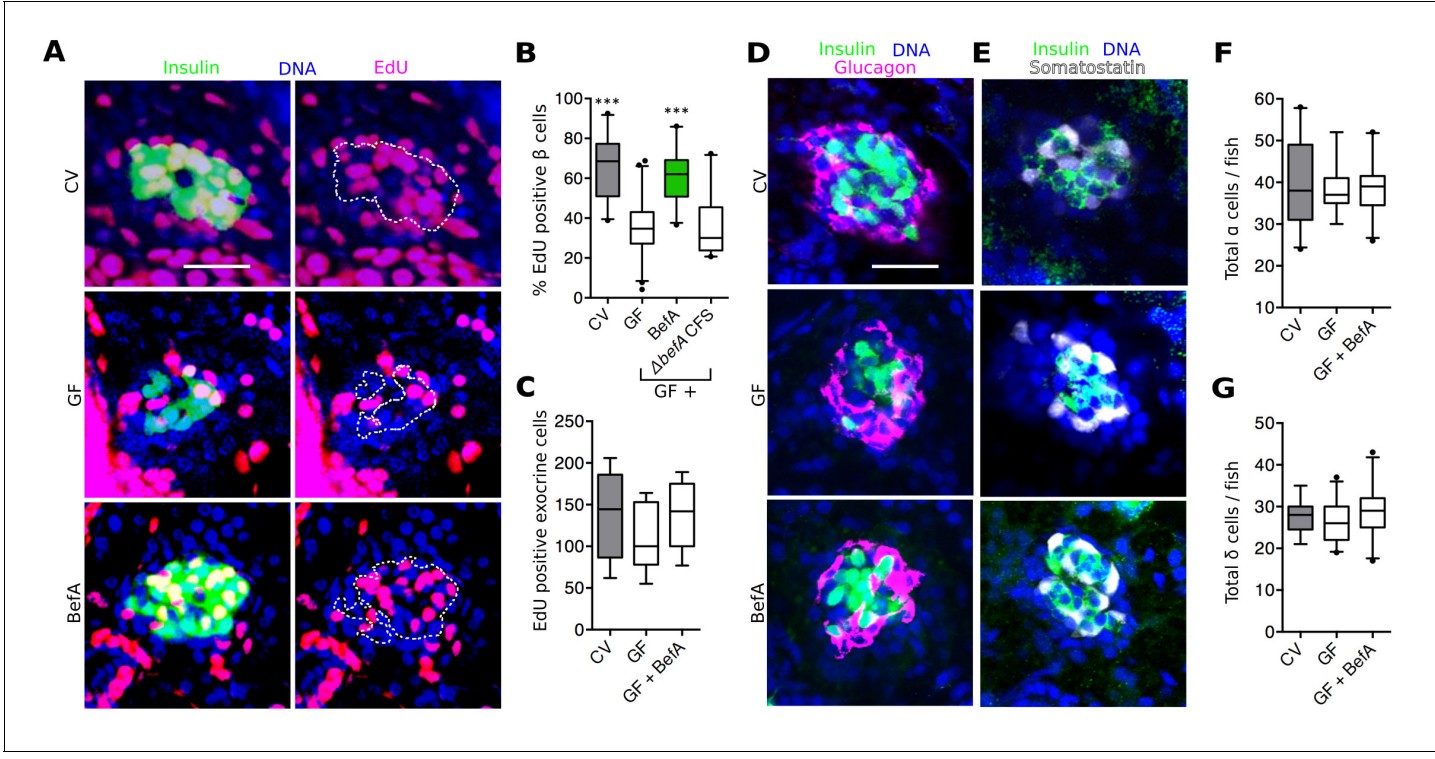

**Figure 5.** BefA facilitates β cell mass expansion through proliferation. (A, D & E) Representative 2D slices from confocal scans through the primary islets of GF, CV and BefA (10165) protein treated 6 dpf larvae. Scale bars = 40 µM. (A) Insulin promoter expressing β cells are shown in green, all nuclei are blue, and EdU containing nuclei are magenta. Left hand panels are a merge of all three markers. For ease of resolving cells that are double positive for both insulin and EdU, the right hand panels show the location of insulin outlined by white dashed lines. (B) Percentage of EdU positive β cells in CV, GF or GF treated with either purified BefA or CFS from *A. veronii*^ΔbefA cultures (△*befA* CFS). ***p<0.001: Denotes treatment that is significantly different than GF by Tukey analysis. (C) Total EdU positive exocrine cells quantified from the approximate central longitudinal plane of the pancreas in each fish. (D) Insulin promoter expressing β cells are shown in green, all nuclei are blue, and α cells, stained with anti-glucagon antibody are magenta. (E) Somatostatin promoter expressing δ cells are shown in white, all nuclei are blue, and β cells stained with anti-insulin antibody are outlined in green. (F) Total α cells in GF, CV and GF fish treated with BefA. (G) Total δ cells in GF, CV and GF fish treated with BefA.

The following source data and figure supplement are available for figure 5:

**Source data 1.** Quantifications and statistical analysis of proliferation of larval β cells corresponding to *Figure 5B*.
**Figure supplement 1.** The microbiota increase β cell neogenesis from the EPD.

*source data 1*). Our results show that BefA is sufficient to increase cell proliferation that gives rise to an expanded β cell population during early larval development. Furthermore, BefA seems to be the only product of the *A. veronii* CFS that is capable of inducing this cell proliferation.

In zebrafish larvae, both the proliferation of existing β cells as well as the proliferation of progenitors contribute to the expansion of β cells that occurs between 4 and 6 dpf (*Dong et al., 2007*; *Field et al., 2003*). Because our 48-hr EdU pulse labeled β cells born from both events, our experiment did not distinguish the exact cell population undergoing proliferation in response to BefA. Due to their low rates of proliferation, dividing β cells are difficult to detect without pulse labeling. Neogenesis of β cells from progenitors is also rare, but can be detected as the appearance of insulin positive cells in the extra-pancreatic duct (EPD) (*Dong et al., 2007*; *Hesselson et al., 2009*). We quantified insulin expressing cells in the EPD in 6 dpf CV and GF larvae. In a survey of over 500 *Tg(-1.0insulin:eGFP)* larvae, we found a slight but significant increase in EPD-localized insulin expressing cells in CV versus GF fish (*Figure 5—figure supplement 1*), suggesting that the microbiota increases endocrine progenitor proliferation. Whether the microbiota also promote proliferation of mature β cells in the islet and whether BefA promotes the proliferation of one or both of these cell populations remains to be determined.

To test whether BefA activity was specific to endocrine tissue, or whether it acts as a nonspecific pro-proliferative stimulant in the pancreas, we analyzed its ability to induce proliferation in exocrine pancreatic tissue by treating *Tg(ptf1a:eGFP)* larvae (*Thisse et al., 2004*) with EdU and BefA from 4 to 6 dpf and quantifying proliferative eGFP positive cells. We found no difference in the level of exocrine cell proliferation across GF, CV and BefA treatments (*Figure 5C*). To test whether β cells were the only endocrine cell type in the islet to be responsive to BefA, we also quantified the total number of glucagon-expressing α (*Figure 5D*) and somatostatin-expressing δ (*Figure 5E*) cells in GF, CV and BefA treated fish. We again found no difference in the total numbers of these cells across treatments (*Figure 5F,G*). These results suggest that in the pancreas, β cells alone are responsive to the presence of BefA.

## BefA homologs are produced by members of the human gut microbiota and have conserved function

We wondered if BefA-like proteins are produced by the human microbiota. Phylogenetic analysis of related sequences in bacterial genomes uncovered close homologs (at least 82% amino acid sequence identity) in many, but not all, species of the *Aeromonas, Vibrio*, and *Photobacterium* genera. We also found an example of a highly related sequence in the human-associated species *Enterococcus gallinarum*, which was likely acquired through a horizontal gene transfer event (*Figure 6A*). Widening the search to include more distant homologs identified potentially related genes in three additional human-associated genera: *Enterobacter, Escherichia*, and *Klebsiella* (*Figure 6B*).

We tested whether representative BefA-like proteins from human-associated bacteria had the capacity to induce β cell expansion in our gnotobiotic zebrafish model. We cloned into BL21 *E. coli* two *befA*-like genes: the more closely related homologue from *Enterococcus gallinarum* and a more distantly related homologue from *Enterobacter aerogenes*. The amino acid sequence alignment of these two homologs against the *Aeromonas* BefA sequence is shown in *Figure 6—figure supplement 1*. Both the *Aeromonas* and *Enterococcus* sequences contain a short N-terminal hydrophobic secretion signal, which is not predicted in the more distant *Enterobacter* sequence. The most conserved region of these proteins is the C-terminal portion, which contains a putative SYLF domain of unknown function. Induction of expression of each gene in *E. coli* yielded CFS that were dominated by each of the respective homologous proteins, in contrast to the CFS from control *E. coli* expressing an empty vector (*Figure 6C*). Upon addition of these supernatants to GF larval zebrafish, we observed rescue of β cell numbers to the CV level with both the *Enterococcus gallinarum* and *Enterobacter aerogenes* proteins, but not the empty vector control (*Figure 6D*, *Figure 6—source data 1*). These results indicate that members of human-associated microbiota produce secreted proteins capable of inducing β cell expansion.

## Discussion

Using a gnotobiotic zebrafish model, we have discovered a class of proteins produced by resident gut bacteria that have the capacity to increase the expansion of pancreatic β cells during early

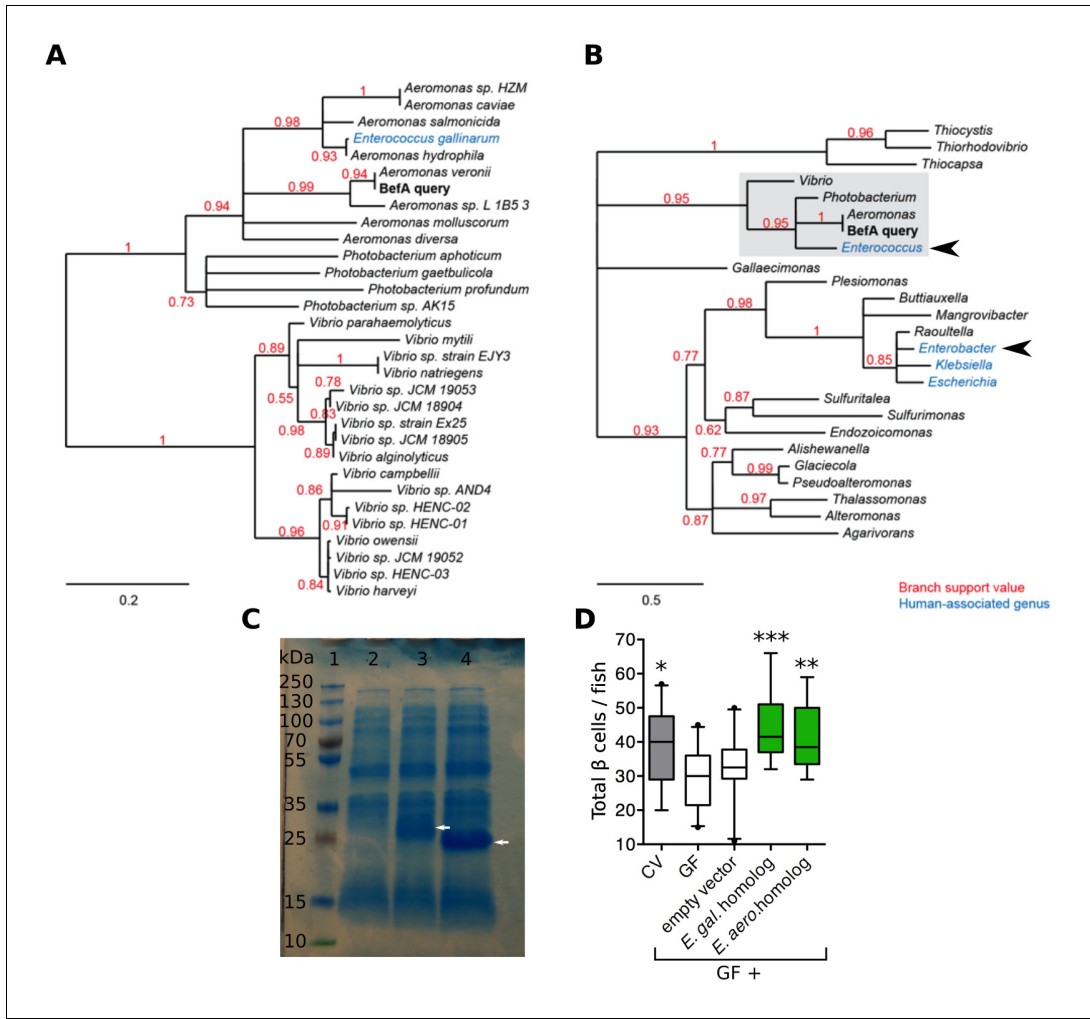

**Figure 6.** Homologs of BefA encoded in the human microbiome have conserved function in zebrafish. (**A**) Close homologs of BefA across microbial species. Each species is represented by its closest BefA homolog, with a minimum allowed amino acid sequence identity of 50% (relative to the query sequence). Notably, the *Enterococcus gallinarum* homolog clusters among homologs from the *Aeromonas* genus, which is evidence of a possible lateral gene transfer event. (**B**) A view of the BefA phylogeny including more distant homologs (sequence identity >20%) and grouped by genus. The portion of the tree represented in **A** is contained in the light gray box. In both panels, red numbers indicate branch support (values closer to 1 are better supported); branches with support values <0.5 have been collapsed. Blue clades indicate genera that were associated with humans in metagenomes produced during the Human Microbiome Project (HMP). Black arrowheads indicate genera tested for functional conservation in panel **D**. Scale bars indicate amino acid substitutions per amino acid site. (**C**) SDS-page gel: 1 = ladder, 2 = CFS from induction of *E. coli* BL21 carrying an empty vector, 3 = CFS from induction of *E. coli* BL21 carrying vector with an *Enterococcus gallinarum* homolog, estimated size of 29 kDa, lane 4 = CFS from induction of *E. coli* BL21 carrying vector with *Enterobacter aerogenes* homolog, estimated size of 21 kDa. White arrows indicate induced proteins. (**D**) Total β cells in CV, GF and GF fish that have been treated with either induced BL21 *E. coli* supernatant dominated by the homologous BefA protein encoded from *Enterococcus gallinarum* (*E. gal.* homolog) and *Enterobacter aerogenes* (*E. aero.* homolog), or induced supernatant from an empty vector control. *p<0.05, **p<0.01, ***p<0.001: Denotes treatment that is significantly different than GF by Tukey analysis.

The following source data and figure supplement are available for figure 6:

**Source data 1.** Quantifications and statistical analysis of larval β cells corresponding to *Figure 6D*.

**Figure supplement 1.** Amino acid sequence alignment of BefA and functionally conserved homologs.

zebrafish development. BefA and related homologues are predicted to contain a C-terminal SYLF domain, which has been described in proteins from organisms in all kingdoms of life, including humans, but for which little is known functionally beyond a possible role in lipid binding (*Hasegawa et al., 2011*). Genes encoding BefA and related proteins are found in a small subset of all bacteria genera, with a predominance in genera of host-associated bacteria, but *befA* homologues are not ubiquitously present in any of these genera.

Our finding of a role for specific secreted bacterial proteins in β cell development raises the possibility of a new link between the resident microbiota and diseases of β cell paucity, such as diabetes mellitus. Type 1 diabetes (T1D), is caused by both genetic and environmental factors, as indicated by the 50% disease discordance among monozygotic twins (*Akerblom et al., 2002*). One environmental factor associated with T1D is microbiota composition (*Gülden et al., 2015*). Mechanistic models for the role of the microbiota in T1D etiology have focused on the capacity of the microbiota to modulate the development and function of the immune system, and thus influence the propensity of genetically susceptible individuals to develop autoimmunity to β cell antigens (*Gülden et al., 2015*). Multiple aspects of host immune cell development and function known to play a role in T1D are altered by the loss of microbes, including development of lymphoid tissue (*Macpherson and Harris, 2004*) and T cell differentiation and function (*Alam et al., 2011*; *Farkas et al., 2015*; *Ivanov et al., 2008*). We hypothesize an additional role for the early microbiota in establishing the β cell population size that would either buffer against, or render individuals susceptible to, β cell depletion by autoimmune destruction.

In humans, β cells undergo a period of postnatal expansion, before becoming quiescent around age two (*Gregg et al., 2012*). Differences in β cell growth during this time are thought to account for the wide variation in β cell mass observed in adults (*Wang et al., 2015b*). The idea that early life β cell census could influence diabetes risk is supported by studies in both rodents and humans, and has been theorized as an important risk factor for type 2 diabetes (*Kaijser et al., 2009*), a disease which is also influenced strongly by microbiota composition (*Cox and Blaser, 2014*). Compromised β cell development in rats results in an insufficient number of cells to adequately control glucose metabolism (*Figliuzzi et al., 2010*). In mice, perinatal β cell proliferation rates can be tuned via the modulation of Gi-GPCR signaling (*Berger et al., 2015*). Changes to early β cell proliferation capacity in these mice correlates directly with adult β cell mass, which subsequently impacts glucose regulation (*Berger et al., 2015*). Furthermore, meta analysis of human data has revealed a correlation between an early age of β cell loss and more rapid onset of T1D (*Klinke, 2008*), consistent with the model that failure to generate a reserve of β cells early in development increases disease risk.

We hypothesize that neonatal microbiomes with a low abundance of BefA equivalents would result in reduced β cell proliferation, lower β cell mass, and increased diabetes risk. We do not know how many different microbiota-derived molecules can stimulate β cell proliferation, but for the case of *befA* homologues, we know these to be sparsely distributed in bacterial genomes, such that microbiomes of low taxonomic diversity could lack these genes. The idea that microbiota-derived factors capable of protecting against diabetes are not widely conserved is consistent with human microbiota profiling data (*Morgan et al., 2013*), our own functional assays of bacteria in gnotobiotic zebrafish, and other gnotobiotic rodent experiments. For example, specific bacterial lineages have been shown to attenuate disease in diabetes models, including *Segmented Filamentous Bacteria* (SFB) in the non-obese diabetic (NOD) mouse (*Kriegel et al., 2011*; *Yurkovetskiy et al., 2013*) and *Lactobacillus johnsonii* in the Biobreeding rat model (*Valladares et al., 2010*). Furthermore, Wen and colleagues have shown that certain microbial assemblages, but not others, confer disease protection in neonatal NOD mice (*Peng et al., 2014*). Additional recent work by Wen and colleagues demonstrates early development as a critical window for microbiota modulation of disease risk in NOD mice (*Hu et al., 2015*). We have shown that BefA acts during early developmental stages in zebrafish, and we hypothesize that β cell expansion during this developmental window is important for disease prevention, and may be a critical period for clinical intervention for infants at risk for T1D development. Further work will be required to determine whether BefA is capable of inducing proliferation of adult β cells in zebrafish or other animals.

Why certain bacteria produce BefA is unclear. In the context of the zebrafish intestinal environment, BefA confers a slight colonization advantage to *A. veronii*, however this is unlikely to be related to its capacity to induce β cell mass, because the colonization requirement is only apparent in the context of co-colonization with wild type *A. veronii* that induce normal β cell numbers. It is

possible that bacterial modulation of host β cell number serves a purpose for the bacteria not measured in our assay. Alternatively, bacteria may produce BefA for a purpose independent of β cell expansion and the host simply uses this bacterial molecule as a cue for its own developmental program. Learning the molecular basis for BefA sensing by the host, and whether it interacts directly or indirectly with β cells, will help shed light on the nature and evolutionary conservation of this interspecies signaling. It will also be important to understand the bacterial function of BefA in order to be able to manipulate its abundance for potential therapeutic purposes.

The incidence of autoimmune diseases such as T1D has been increasing markedly in developed nations over the past several decades. One theory to explain this phenomenon is the disappearing microbiota hypothesis, which proposes that over time, as our modern lifestyles have become increasingly sterile, we have lost ancestral microbial symbionts important for specific aspects of our health (*Blaser and Falkow, 2009*). Our discovery of a specific class of bacterial proteins that promote β cell expansion in early development is consistent with the hypothesis that loss of specific microbial taxa from gut microbiota could underlie increased diabetes risk. Specifically, we suggest that BefA-like proteins promote the establishment of a robust β cell population that is more resilient to subsequent β cell loss. Because *befA* is a relatively rare component of the microbiome, we cannot measure it directly from available metagenomic sequence data to test our hypothesis that *befA* abundance correlates with reduced diabetes risk. The low abundance of *befA* in metagenomes also highlights the challenge of discovering disease determinants from metagenomic data, and emphasizes the importance of functional screening approaches. The larval zebrafish has served as a valuable high-throughput vertebrate model for the identification of new compounds and pathways that can increase β cell numbers exogenously (*Andersson et al., 2012*; *Wang et al., 2015a*). We have employed the gnotobiotic zebrafish to explore how microbial cues modulate β cell development. Our discovery of BefA highlights the importance of the microbiota in shaping the development of an extra-intestinal tissue and influencing the overall metabolic state of the host. We postulate that resident bacteria are a rich and underexplored source of functionally conserved molecules that shape early host development in ways that impact disease risk in later life.

## Material and methods

### Gnotobiotic zebrafish

All zebrafish experiments were performed using protocols approved by the University of Oregon Institutional Care and Use Committee and followed standard protocols. Zebrafish embryos were derived germ-free (GF) as previously described (*Bates et al., 2006*). XGF and mono-associated larvae were also generated as previously described (*Bates et al., 2006*), except that all bacterial inoculate were added to GF flasks at 4 dpf at a final concentration of $10^6$ CFUs/mL. In experiments quantifying the colonization levels of bacterial isolates, each strain was added to the embryo media (EM) and incubated with the larvae for 48 hr at 27°C. Larvae were sacrificed at 6 dpf, immediately before the gut was removed and homogenized in a small sample of sterile EM. Dilutions of this gut slurry were plated onto tryptic soy agar and allowed to incubate overnight at 30°C. Colonies from each gut were quantified. A minimum of 10 guts per mono-association or di-association were analyzed.

### Free glucose assay

To measure β cell function in GF and CV zebrafish larvae, levels of free glucose were measured at 6 dpf using a free glucose assay kit (BioVision, Milpitas, CA) as described previously (*Andersson et al., 2012*; *Gut et al., 2013*) except that only 10 larvae were combined per tube. Three to five biological replicates (sets of 10 larvae) were completed for both GF and CV treatments each time the assay was conducted. Data shown here were combined from 3 separate experimental assays or technical replicates.

### Cell free supernatant

GF fish were inoculated with secreted bacterial products at 4 dpf by adding cell free supernatant (CFS) at a final concentration of 500 ng/mL to the water of the sterile flasks. CFS was harvested from a 50 mL overnight culture of the specified bacterial strain. The cultures were centrifuged at 7000 g

for 10 min at 4°C. The supernatant was then filtered through a 0.22-µm sterile tube top filter (Corning Inc., Corning, NY); sterile supernatant was concentrated at 4°C for 1 hr at 3000 g with a centrifugal device that has a 10 kDa weight cut off (Pall Life Sciences, Port Washington, NY).

For experiments utilizing proteinase K (Qiagen, Hilden, Germany), the enzyme was added to samples of CFS at a final concentration of 100 µg/mL and allowed to incubate at 55°C for 1 hr before inactivating the enzyme at 90°C for 10 min.

## Ammonium sulfate fractionation

Ammonium sulfate fractionation was performed on un-concentrated, sterile CFS from a 50 mL overnight culture by slowly adding 100% ammonium sulfate until solutions of 20%, 40%, 60% and 80% ammonium sulfate were achieved. These solutions were prepared at 4°C. Precipitated proteins were collected from each fraction by centrifugation at 4°C and 14,000 g for 15 min. The proteins were resuspended in cold EM and dialyzed for 2–3 hr at 4°C before adding them to 4 dpf GF larvae at a final concentration of 500 ng/mL.

## Mass spectrometry

The 60–80% ammonium sulfate fraction of the *A. veronii*$^{\Delta T2SS}$ CFS was sent to the Proteomics Lab at Oregon Health and Science University in Portland, OR for protein identification (partial sequencing) analysis.

## Protein expression and purification

The nucleotide sequence for the *befA* gene from was amplified from *A. veronii* using the following forward and reverse PCR primers respectively: 5'-GCCCATATGatgaacaagcgtaactggttgctg-3' and 5'-GGCCTCGAGgcggctcgtttcagtcaagtc-3'. The nucleotide sequences for both the *Enterococcus gallinarum* and *Enterobacter aerogenes befA* gene homologs were obtained from NCBI and subsequently synthesized by GenScript, Piscataway, NJ. Each of these two genes was then cloned separately into the pET-21b plasmid (Novagen, Darmstadt, Germany), which contains an IPTG inducible promoter. A His•Tag was added to the C-terminal of the original BefA protein sequence for subsequent purification. As a control, a second version was also constructed lacking the tag. These vectors were then transformed into BL21 *Escheria coli* (RRID:WB_HT115(DE3)), treated with 0.5 – 1.0 mM IPTG during exponential growth phase (OD$_{600}$ = 0.4–0.6) and allowed to grow for 3–4 more hours at 30°C. This resulted in both a CFS and cell lysate dominated by our proteins of interest, as confirmed via SDS-page gel electrophoresis by the presence of dark bands of the expected sizes for each protein. These bands were absent from BL21 cultures carrying an empty pET-21b vector. The CFS from these inductions was added to GF zebrafish at 4 dpf at a final concentration of 500 ng/mL.

For purification of BefA, IPTG induced BL21 cells were sonicated at 32,000 g in a 50 nM Tris, 150 mM NaCl buffer (buffer A). The supernatant was then added to a solution of nickel beads (Thermo Scientific HisPur Ni-NTA Resin, Waltham, MA) to capture the His•tag. The beads were washed several times in a 30 mM imidazol solution in buffer A and subsequently eluted in 300 mM imidazole solution in buffer A. The isolation of pure BefA was confirmed with SDS-page gel electrophoresis by the presence of a single band of about 29 kDa in size. Purified BefA was added to 4 dpf GF fish at a final concentration of 500 ng/mL.

## Experimental bacterial strains

To create the *A. veronii*$^{\Delta befA}$ mutant strain, a vector containing a chloramphenicol resistance cassette was transformed into SM10 *E. coli*. Conjugation between wild-type *Aeromonas veronii* HM21 and the vector carrying SM10 *E. coli* strain was carried out, allowing the chloramphenicol resistance gene to replace the *befA* locus in *A. veronii* via allelic exchange. Candidate mutants were selected for loss of the plasmid and maintenance of chloramphenicol resistance. Insertion of the chloramphenicol cassette into the *befA* locus was verified in these candidates by PCR. Joerg Graf graciously provided us with the *A. veronii*$^{\Delta T2SS}$ strain (**Maltz and Graf, 2011**).

## Primary islet cell type quantifications and EdU staining in larvae

*Tg(-1.0insulin:eGFP)* (RRID:ZFIN_ZDB-GENO-100513-10, ZIRC, Eugene, OR) (**diIorio et al., 2002**) zebrafish embryos were used to visualize and quantify the total number of β cells in developing

larvae. *Tg(insulin:PhiYFP-2a-nsfB, sst2:mCherry)* (RRID:ZFIN_ZDB-GENO-120217-6) (*Wang et al., 2015a*) were obtained from Jeff Mumm and were used to visualize and quantify δ cells. All experiments were analyzed at 6 dpf unless otherwise specified. At all time points in all experiments, larvae were fixed with 4% paraformaldehyde supplemented with 0.01% TritonX-100 (Thermo Fisher Scientific, Waltham, MA) at 4°C overnight, or at room temperature for 2–3 hr, and then washed with PBS. The following antibodies were used to distinguish α and β cells: guinea-pig anti-insulin (Dako Cat# A0564, RRID:AB_10013624, Carpinteria, CA), mouse anti-glucagon (Sigma-Aldrich Cat# G2654, RRID:AB_259852), St. Louis, MO), rabbit anti-GFP (Molecular Probes Cat# A-11122, RRID:AB_ 221569), mouse anti-mCherry (Abcam Cat# ab125096, RRID:AB_11133266, Cambridge, MA), Alexa Fluor 488 goat anti-rabbit (Thermo Fisher Scientific, Waltham, MA), anti-mouse Cy3 (Jackson ImmunoResearch Laboratories Inc., West Grove, PA), Alexa Fluor 488 goat anti-guinea-pig (Thermo Fisher Scientific, Waltham, MA), and TO-PRO-3-Iodide (642/661) (Thermo Fisher Scientific, Waltham, MA).

For experiments quantifying proliferation, EdU was added at 4 dpf directly to the EM at a final concentration of 0.1 mg/mL. The Click-iT EdU Imaging Kit (Thermo Fisher Scientific, Waltham, MA) was used to process the EdU label in whole fixed zebrafish prior to antibody staining, according to the manufacturer's protocols. Whole, antibody-stained larvae were mounted for confocal microscopy (BioRad Radiance 2100) with their right side facing up against the cover slip, which was flattened sufficiently to spread out the cells within the islet for optimal quantification of individual cells. For quantification of β cells and other primary islet cells, the entire endocrine portion of the pancreas was scanned using a 60X objective (Nikon Eclipse E600FN), and Fiji (RRID:SCR_002285) (*Schindelin et al., 2012*) software was used to analyze each image stack. For quantification of pancreatic exocrine tissue proliferation, *Tg(ptf1a:eGFP)* (RRID:ZFIN_ZDB-GENO-080111-1, ZIRC, Eugene, OR) (*Thisse et al., 2004*) zebrafish were scanned through the entire pancreas with a 20X objective (Nikon Eclipse E600FN) and Fiji was used to analyze the percentage of proliferative cells in single sections from the center of the organ. Images were prepared for publication using the open source Inkscape software (RRID:SCR_014479).

For experiments quantifying insulin-expressing cells in the region of the EPD, zebrafish were processed as described above, and analyzed on a Leica fluorescent microscope using a 2x objective.

## BefA phylogenetic analysis

We screened for BefA homologs across microbial species using a *blastp*-based (*Altschul et al., 1997*) search of the UniProt Knowledgebase (*UniProt Consortium, 2015*) (version 6/2015); default search parameters were changed to allow (i) a maximum *E*-value of 1.0 and (ii) an arbitrarily large number of database hits. We classified database hits as 'close homologs' if amino acid sequence identity exceeded 50% (relative to the query length) and 'distant homologs' if their percent identity exceeded 20%. For phylogenetic analysis at the species level, each species was represented by the hit of highest percent identity to BefA among isolates of that species (if any); an analogous procedure was used for genus-level analysis. Aligned portions of database sequences were isolated and multiply aligned with *MUSCLE* (RRID:SCR_011812) (*Edgar, 2004*). Phylogenetic trees were constructed from these multiple sequence alignments using *PhyML* (RRID:SCR_014629) (*Guindon and Gascuel, 2003*) and visualized within the *Phylogeny.fr* webserver (*Dereeper et al., 2008*). Microbial genera were classified as 'human-associated' if they occurred with relative abundance >0.01% in at least 5 metagenomes from the Human Microbiome Project (*Huttenhower et al., 2012*) as profiled by MetaPhlAn (RRID:SCR_004915) (*Segata et al., 2012*). Secretion signal peptides were predicted from amino acid sequences using SignalP (*Petersen et al., 2011*).

## Statistical analysis

Appropriate sample sizes for all experiments were estimated *a priori* using a power of 80% and a significance level of 0.05. From preliminary experiments we estimated variance and effect. For larval β cell quantification, these parameters suggested using a sample size of 30 in order to detect significant changes between treatment groups. Therefore, each experiment contained about 10–15 biological replicates or individual fish per treatment group, although some larger experiments had fewer biological replicates due to limited material. Entire experiments or technical replicates were repeated multiple times, resulting in pooled data sets of about 20–50 biological replicates. These data are represented in the figures as box and whisker plots, which display the data median (line

within the box), first and third quartiles (top and bottom of the box), and 95% confidence interval (whiskers). Any data point falling outside the 95% confidence interval is represented as a solid dot. These pooled data were analyzed through the statistical software RStudio. For experiments comparing just two differentially treated populations, a Student's t-test with equal variance assumptions was used. For experiments measuring a single variable with multiple treatment groups, a single factor ANOVA with post hoc means testing (Tukey) was utilized. A p-value of less than 0.05 was required to reject the null hypothesis that no difference existed between groups of data.

## Acknowledgements

We thank J Graf for providing us with the *Aeromonas veronii* HM21strains, T Wiles for guidance in the construction of the *Aeromonas veronii* HM21$^{\Delta befA}$ strain, A Rolig and A Banse for technical advice with CFS preparation, E Goers Sweeny for technical advice with protein purification, P De Verteuil for assistance with colonization experiments, R Sockol and the University of Oregon Zebrafish Facility for maintenance of zebrafish lines, and JPostlethwait, J Eisen, A Powell, A Rolig and T Jones for critical reading of the manuscript. Research reported in this publication was supported by the National Institute of General Medical Sciences of the National Institutes of Health under award numbers P50GM098911 and T32 GM007413-37. Grant P01HD22486 provided support for the Oregon Zebrafish Facility. The content is solely the responsibility of the authors and does not necessarily represent the official views of the NIH.

## Additional information

### Funding

| Funder | Grant reference number | Author |
|---|---|---|
| National Institutes of Health | P50GM098911 | Karen Guillemin |
| National Institutes of Health | T32 GM007413-37 | Jennifer Hampton Hill |
| National Institutes of Health | P01HD22486 | Karen Guillemin |

The funders had no role in study design, data collection and interpretation, or the decision to submit the work for publication.

### Author contributions

JHH, Conception and design, Acquisition of data, Analysis and interpretation of data, Drafting or revising the article; EAF, CH, Acquisition of data, Analysis and interpretation of data; KG, Conception and design, Analysis and interpretation of data, Drafting or revising the article

### Author ORCIDs

Karen Guillemin, http://orcid.org/0000-0001-6004-9955

### Ethics

Animal experimentation: This study was performed in strict accordance with the recommendations in the Guide for the Care and Use of Laboratory Animals of the National Institutes of Health. All zebrafish experiments were performed using protocols approved by the University of Oregon Institutional Care and Use Committee under protocol number 15-83 and institutional Animal Welfare Assurance number A-3009-01.

## Additional files

### Supplementary files

• Supplementary file 1. Mass spectrometry: *Aeromonas* CFS inventory. Relative abundance of protein species (spectral count) within the 60–80% ammonium sulfate fraction of the *A. veronii*$^{\Delta T2SS}$ CFS. BefA is listed in bold and highlighted in green (uncharacterized protein).

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
