## [Decision Letter]

Thank you for submitting your article "A conserved bacterial protein induces pancreatic beta cell expansion during zebrafish development" for consideration by *eLife*. Your article has been favorably evaluated by Marianne Bronner as the Senior Editor and three reviewers, including Daniel Hesselson (Reviewer #2) and a member of our Board of Reviewing Editors.

The reviewers have discussed the reviews with one another and the Reviewing Editor has drafted this decision to help you prepare a revised submission.

Summary:

In this elegant paper from the Guillemin lab, the authors report the surprising discovery of a bacterial protein that positively regulates the number of pancreatic β cells in zebrafish by inducing β cell proliferation. The data are in general of high quality (see below for some concerns) as is the writing, and the significance of the findings is very high.

Essential revisions:

1) The variability of the wild-type sample in Figure 1 (especially at 6 dpf) is really quite surprising (previous publications have not reported such variability) and could be due to a number of factors including the method used to count (transgene vs. Ab), the imaging platform, the counting method itself (by hand vs using a particular software), whether the fish are well synchronized and healthy (and thus with a similar amount of yolk left), and whether they are fed or not(the variability in control animals in the later figures appears to be reduced).

[Ideally, the combination of a membrane and a nuclear marker would be best.]

It is also possible that the variability in Figure 1 comes from non-random sampling of the cohort. If the authors started with a pool of animals at 3 dpf and chose the healthiest/most developed larvae each day then collected the remainder at 6 dpf that would explain the apparent loss of β cells and increased variability between 5 dpf and 6 dpf in the CV controls. Perhaps this experiment should be repeated with a defined sampling plan.

2) In the germ free animals (e.g., Figure 1), what is the identity of the other cells in the islet? The authors should stain the animals for glucagon and somatostatin to determine A) whether CV vs GF conditions affect differentiation of the other endocrine cell types and B) if α/δ cell mass is affected whether this activity is also mediated by BefA. It is also possible that BefA affects hormone expression rather than cell differentiation which would also be an interesting observation.

3) Does the microbiota/BefA act solely by stimulating proliferation of mature β cells or does it also have effects on endocrine progenitors? In mammals these processes are temporally separated with the vast majority of neogenesis occurring before birth under sterile conditions followed by diet (and now possibly microbiota) induced proliferation postnatally. In zebrafish both of these processes occur concurrently during early larval development and contribute to the expansion of the β cell mass from 3 to 6 dpf. The 48 h EdU pulse used in Figure 5 would have labeled proliferating endocrine progenitor cells that subsequently differentiated into β cells in addition to proliferating β cells. Phospho-histone H3 staining should be used to clarify whether BefA stimulates proliferation of mature β cells.

---

## [Author Response]

*[…] Essential revisions:*

*1) The variability of the wild-type sample in Figure 1 (especially at 6 dpf) is really quite surprising (previous publications have not reported such variability) and could be due to a number of factors including the method used to count (transgene vs. Ab), the imaging platform, the counting method itself (by hand vs using a particular software), whether the fish are well synchronized and healthy (and thus with a similar amount of yolk left), and whether they are fed or not (the variability in control animals in the later figures appears to be reduced).*

*[Ideally, the combination of a membrane and a nuclear marker would be best.]*

*It is also possible that the variability in Figure 1 comes from non-random sampling of the cohort. If the authors started with a pool of animals at 3 dpf and chose the healthiest/most developed larvae each day then collected the remainder at 6 dpf that would explain the apparent loss of β cells and increased variability between 5 dpf and 6 dpf in the CV controls. Perhaps this experiment should be repeated with a defined sampling plan.*

We are confident that the fish used for Figure 1 were controlled with respect to health, developmental synchronization, feeding (all fish are starved for GF derivation protocols), and experimental method. The data were also randomly sampled, as we raised a separate pool of fish for each day samples were taken.

The data in our manuscript is displayed using box plots, with whiskers denoting 5-95% of the values of the *entire* data set, whereas the data from other manuscripts quantifying β cell number is most often displayed using a histogram, with error bars representing either the standard error (SE) or the standard error of the mean (SEM) (Andersson et al. Cell Met. 2012, Hesselson et al. *PNAS*. 2009, Wang et al. *eLife*. 2015, Ye et al. Development. 2015). We chose to use box plots to represent our data to convey maximal information and transparency, as discussed by Krzywinski and Altman in Nature Methods in 2014. It is difficult to get an indication of the total variation of a data set from SE and SEM. These measurements are usually much smaller than the total variation we have shown in our box plots. Therefore, it is understandable that our figures appear to have larger than normal variation. For comparison, we transformed Figure 1 into a histogram with error bars representing SEM, and placed it next to a similar graph (Figure 3, panel G) from Andersson et al. Cell Met. 2012, which also used SEM. Of note, the error bars from both works are similar in size. The exact numerical values for the standard deviation as well as the SEM for all of our data can also be found in the corresponding source data file. Descriptions of our graphical representations are also indicated in our manuscript within the figure legends and methods.

*2) In the germ free animals (e.g., Figure 1), what is the identity of the other cells in the islet? The authors should stain the animals for glucagon and somatostatin to determine A) whether CV vs GF conditions affect differentiation of the other endocrine cell types and B) if α/δ cell mass is affected whether this activity is also mediated by BefA. It is also possible that BefA affects hormone expression rather than cell differentiation which would also be an interesting observation.*

We have added quantifications of both glucagon-expressing α and somatostatin-expressing δ cells in GF, CV and BefA treated larvae to Figure 5. Text descriptions of these results start in the last paragraph of the subsection “BefA facilitates β cell expansion by inducing proliferation”. We found no change in the total numbers of either of these cell types across treatments.

*3) Does the microbiota/BefA act solely by stimulating proliferation of mature β cells or does it also have effects on endocrine progenitors? In mammals these processes are temporally separated with the vast majority of neogenesis occurring before birth under sterile conditions followed by diet (and now possibly microbiota) induced proliferation postnatally. In zebrafish both of these processes occur concurrently during early larval development and contribute to the expansion of the β cell mass from 3 to 6 dpf. The 48 h EdU pulse used in Figure 5 would have labeled proliferating endocrine progenitor cells that subsequently differentiated into β cells in addition to proliferating β cells. Phospho-histone H3 staining should be used to clarify whether BefA stimulates proliferation of mature β cells.*

As suggested, we performed anti-PH3 antibody staining on 6 dpf larvae to determine whether BefA or other microbes could increase proliferation of mature β cells. We found PH3 marked β cells to be very rare in all treatments. Based on the initial results from our experiment of about 50 fish per treatment, we calculated the number of samples needed to adequately power a study that *would*be able to detect a difference in PH3 staining between our treatments. We approximated that 1300 fish per treatment were required to be confident that no difference actually exists in PH3 staining. We are unable to perform an experiment of this scale and therefore we cannot support or refute the possibility that BefA acts by inducing proliferation of existing β cells.

In parallel, we measured the effect of the microbiota on neogenesis. Since differentiation of β cells from 4-6 dpf occurs within the extra-pancreatic duct (EPD) (Dong et al., 2007; Hesselson et al., 2009), we quantified the number of insulin positive cells located in the region of the EPD in 6 dpf GF and CV larvae. Again, detection of β cells in this region at a single moment in time is a relatively rare event, but we were able to use larvae from past experiments to gather sufficient sample sizes to power this study. We found that there were significantly more EPD localized insulin+ cells in CV fish than in GF fish. We have included a supplemental figure (Figure 5—figure supplement 1) with these data. We also quantified EPD localized β cells in BefA treated fish, however we did not have a sufficient sample size (which we calculated to be about 200 fish) to detect a statistically significant difference, although the result is trending in the direction of CV larvae.

Unfortunately, we cannot conclusively determine the mechanism by which BefA increases EdU incorporation in β cells. We have changed the text to reflect the possibilities discussed above, in the subsection “BefA facilitates β cell expansion by inducing proliferation”. We believe that this question is important, and should be more extensively explored in a future publication focused on how the host detects and interprets BefA.